# Genomic surveillance of SARS-CoV-2 Spike gene by sanger sequencing

**Tiago Souza Salles**[1][⚬], **Andrea Cony Cavalcanti**[1,2][⚬], **Fábio Burack da Costa**[1], **Vanessa Zaquieu Dias**[1], **Leandro Magalhães de Souza**[2], **Marcelo Damião Ferreira de Meneses**[1], **José Antônio Suzano da Silva**[1,3], **Cinthya Domingues Amaral**[1], **Jhonatan Ramos Felix**[1,4], **Duleide Alves Pereira**[1], **Stefanella Boatto**[3], **Maria Angélica Arpon Marandino Guimarães**[5], **Davis Fernandes Ferreira**[1], **Renata Campos Azevedo**[1]*

**1** Department of Virology, Institute of Microbiology Paulo de Góes, Federal University of Rio de Janeiro, Rio de Janeiro, Brazil, **2** Central Laboratory of Public Health Noel Nutels - LACEN-RJ, Rio de Janeiro, Brazil, **3** Department of Applied Mathematics, Institute of Mathematics, Federal University of Rio de Janeiro, Rio de Janeiro, Brazil, **4** Department of Mathematics, Institute of Mathematics, Federal University of Rio de Janeiro, Rio de Janeiro, Brazil, **5** Department of Preventive Medicine, Hospital Universitário Clementino Fraga Filho, Federal University of Rio de Janeiro, Rio de Janeiro, Brazil

⚬ These authors contributed equally to this work.
* renatacampos@micro.ufrj.br

**Data Availability Statement:** This methodology covered 100% of the S gene sequenced (3,822 pb). The sequences obtained were deposited at GISAID (numbers EPI_ISL_4496739,

## Abstract

The SARS-CoV-2 responsible for the ongoing COVID pandemic reveals particular evolutionary dynamics and an extensive polymorphism, mainly in Spike gene. Monitoring the S gene mutations is crucial for successful controlling measures and detecting variants that can evade vaccine immunity. Even after the costs reduction resulting from the pandemic, the new generation sequencing methodologies remain unavailable to a large number of scientific groups. Therefore, to support the urgent surveillance of SARS-CoV-2 S gene, this work describes a new feasible protocol for complete nucleotide sequencing of the S gene using the Sanger technique. Such a methodology could be easily adopted by any laboratory with experience in sequencing, adding to effective surveillance of SARS-CoV-2 spreading and evolution.

## Introduction

The SARS-CoV-2 responsible for atypical pneumonia, evidenced in China by the end of 2019, was classified into the severe acute respiratory syndrome-related coronaviruses, member of *Betacoronavirus* genus, *Coronaviridae* family, been denominated Severe Acute Respiratory Syndrome Coronavirus 2 (SARS-CoV-2).

Coronaviruses are enveloped positive single-strand RNA viruses, with 30,000 bases in length, being the largest RNA genome identified up to date [1]. The SARS-CoV-2 genome has several ORFs; the first ORF1a/b stands at the RNA 5' end and translates the non-structural proteins (nsP1 –nsP16). The RNA 3' end holds the genes of the four structural (E, M, N e S) and accessories proteins. In the mature virus particle, protein S, a homo-trimeric type I fusion glycoprotein, is located on the surface of the virus particle and is responsible for binding to the

EPI_ISL_4497141, EPI_ISL_4497286) and
GenBank (numbers OM064632, OM064633,
OM064634).

**Funding:** This work was material supported by
Fundação Carlos Chagas Filho de Amparo à
Pesquisa do Estado do Rio de Janeiro-FAPERJ
[grant number E-26/201.840/2017] (RCA) and
Coordenação de Aperfeiçoamento de Pessoal de
Nível Superior - Brasil (CAPES) – Public Notice
Number 09/2020 - Prevention and Combat against
Outbreaks, Endemics, Epidemics and Pandemics.
Process number 223038.014313/2020-19 (TSS
and FBC). The funders had no role in study design,
data collection and analysis, decision to publish, or
preparation of the manuscript.

**Competing interests:** The authors have declared
that no competing interests exist.

cell receptor. In humans, the angiotensin-converting molecule (ACE-2) was assigned as the primary receptor for SARS-CoV2.

Several research groups have solved the complete structure of the SARS-CoV-2 S protein attached or not to the receptor ACE-2 [2]. This protein has approximately 1,273 amino acids, and its domains are delimited. Due to the relevance for virus attachment and entrance at susceptible cells, mutations in the receptor-binding domain (RDB) receive greater attention. In addition, mutations at other domains, like the amino (N) -terminal domain (NTD), can also lead to conformational changes in S protein structure and impact their function [3].

SARS-CoV-2 has particular evolutionary dynamics, and an extensive polymorphism is observed. However, the frequency of mutation across the SARS-CoV-2 genome is not uniform. Polymorphism (SNP) is mainly observed in protein S, RNA polymerase, RNA primase, and nucleoprotein [4]. According to the World Health Organization (WHO), isolates those present changes in amino acids that lead to suspected or confirmed cases with a phenotypic impact are considered variants of interest (VOI). Furthermore, these variants are classified as a concern (VOC) when they are associated with increased transmissibility, virulence, changes in the clinical presentation of COVID-19, and reduced containment measures, such as escaping diagnostic tools decreasing the effectiveness of vaccines and therapies [5].

Since the S protein is the primary target of neutralizing antibodies, monitoring insertions, deletions, or substitutions of amino acids can reveal variants with the potential to evade vaccine immunity. In this context, genomic information is quickly shared through initiatives like the GISAID platform, and variants are counted and georeferenced [6]. Up to Jun 2021, five variants were classified as a concern (VOC), named; B.1.1.7 (Alpha), B.1.351(Beta), P.1 (Gamma), B.1.617+ (Delta), first detected in the United Kingdom, South Africa, Brazil, and India, respectively (Fig 1).

Early identification of the variants of concern (VOC) could provide excellent auxiliary information to decision making, allowing an earlier action towards measures to refrain the spreading of the virus such as reinforcement of mobility restriction or relaxation of such measures in areas where the variants are no present. Fig 2 shows a great variety in COVID-19 lethality in the different countries around the world. Unfortunately, due to the imposing genome size of the SARS-CoV-2, economic and laboratory challenges are manifest when monitoring the evolution of this virus. Fig 3 exhibits the significant disparity in the genome shared distribution per country.

Despite the reduction in the costs of new generation sequencing (NGS), the implementation of this system still requires a significant financial contribution, and the price per sample remains high for developing countries. The discrepancy in the number of sequences deposited in databases between countries reflects the difficulties of sequencing, as also shown in Table 1.

Unlike NGS methodologies, nucleotide sequencing based on the Sanger technique is widespread worldwide. In addition, the costs for sequencing small fragments are affordable. Therefore, to support the urgent surveillance of changes in SARS-CoV-2 S gene, this work describes a feasible protocol for complete nucleotide sequencing of the S gene using the Sanger technique. Thus, any laboratory with experience in sequencing can adopt this protocol.

## Materials & methods

### Ethics and study population

This work was previously approved by the Ethics Committee of Clementino Fraga Filho University Hospital (HUCFF/UFRJ) (number: 4.546.307). To evaluate this study, three samples from patients of confirmed COVID-19 presenting high viral load (Ct value < 20) were randomly selected. Patients were admitted to different hospitals in Rio de Janeiro, and a

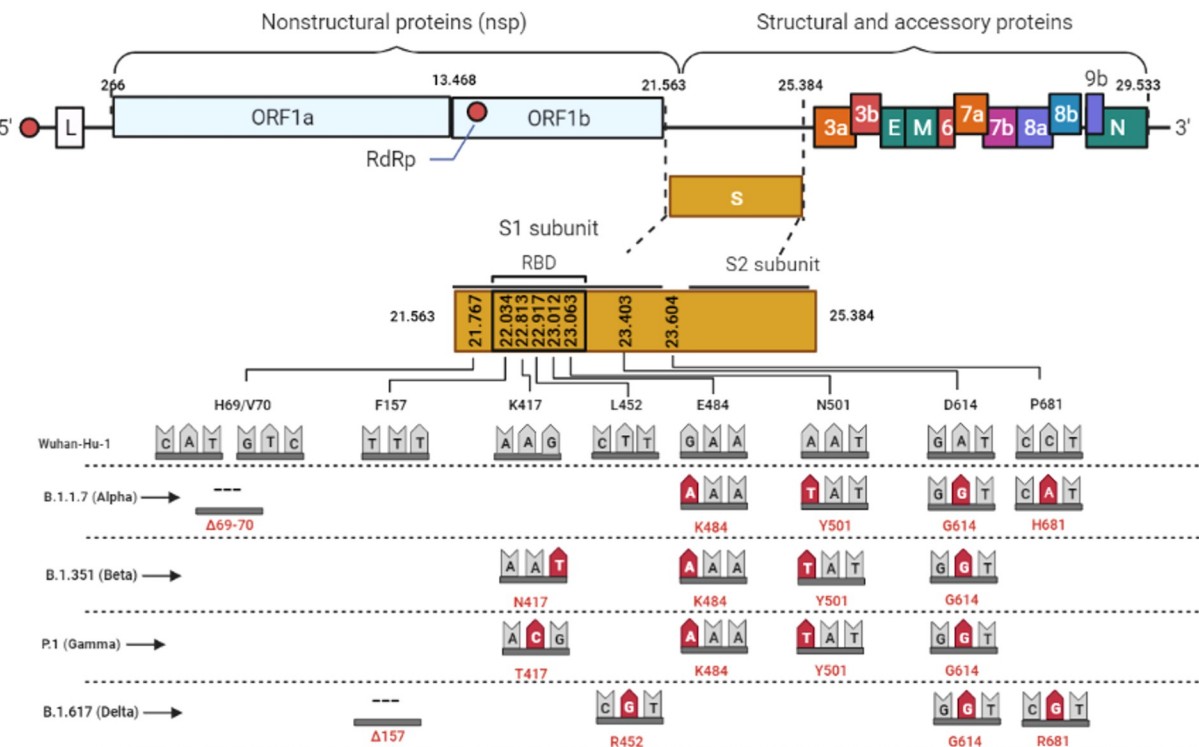

**Fig 1. Graphical representation of the SARS-CoV-2 genome.** Highlighted are the S gene and the main mutations described in the variants of concern. As a measure of comparison, the length of the S gene is already equivalent to the one of the whole Dengue virus genome.

nasopharyngeal swab was collected to confirm clinical diagnosis by Rio de Janeiro Public Health Reference Laboratory—LACEN-RJ. Human samples were used after the conclusion of the diagnostic investigation. All patients' personal information was anonymized, only the municipalities of residence were disclosed. Therefore, the ethics committee waived the requirement for informed consent from patients.

## RNA extraction

According to the manufacturer's instructions, the commercial kit MagMax Viral Pathogen (Thermo fisher, EUA) was used in the automated equipment King Fisher Apex (Thermo Fisher, EUA) to obtain the viral RNA from 200uL of respiratory secretion samples collected in nasopharyngeal swabs.

## RT-qPCR for detection of SARS-CoV-2

The suspected samples of COVID-19 were tested in the diagnostic routine of the Noel Nutels Central Public Health Laboratory (LACEN-RJ) using the SARS-CoV-2 Duplex Kit (E/RP), Biomaguinhos (Fiocruz, Brasil). The reactions were performed using the QuantStudio 5 (Applied Biosystems, Thermo Fisher, EUA). The samples with ct values below 20 were selected for sequencing.

## Amplification of the S protein gene

Six sets of primers targeting the S segment and two sets flanking it were designed based on the sequences deposited in GISAID until September 2020. 29 samples from 13 regions were

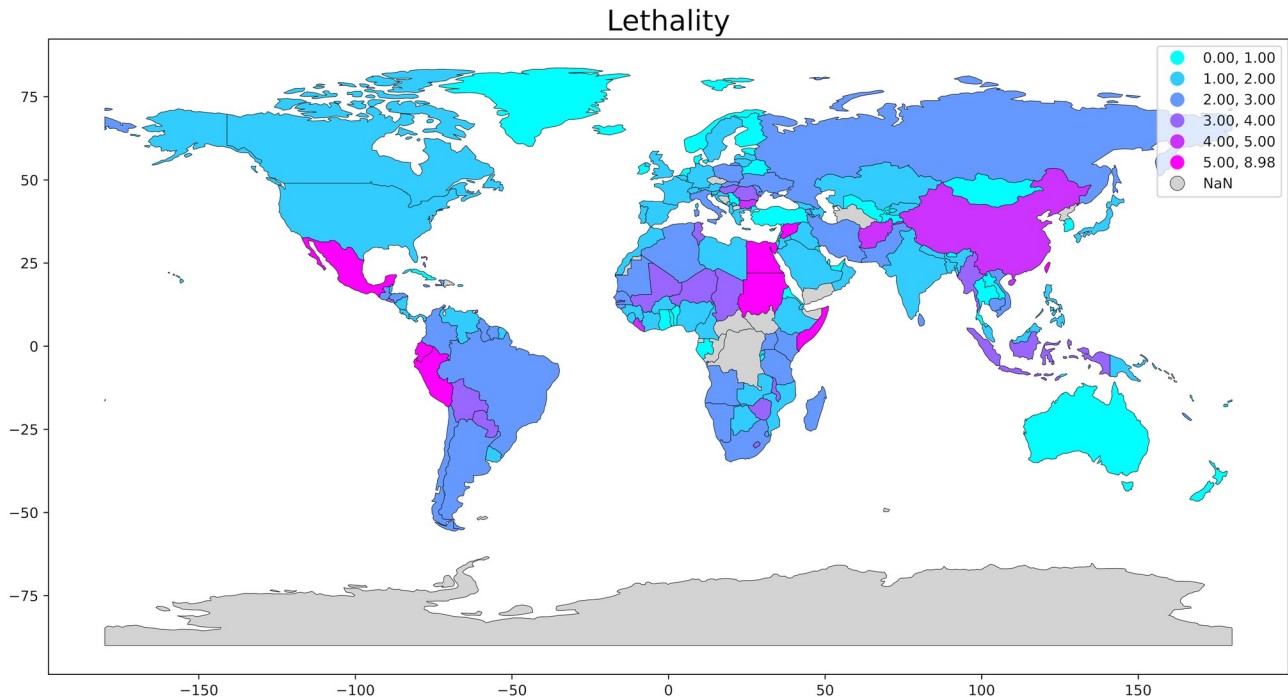

**Fig 2. Lethality [(Reported COVID-19 deaths)/(Reported COVID-19 cases)] per country.** We made use of the GISAID platform's data to estimate COVID lethality and genome sharing per $10^5$ inhabitants. We obtained the geospatial data for plotting the map in the open-source software library written for the Python programming language, Geopandas. The areas in grey are without reported data.

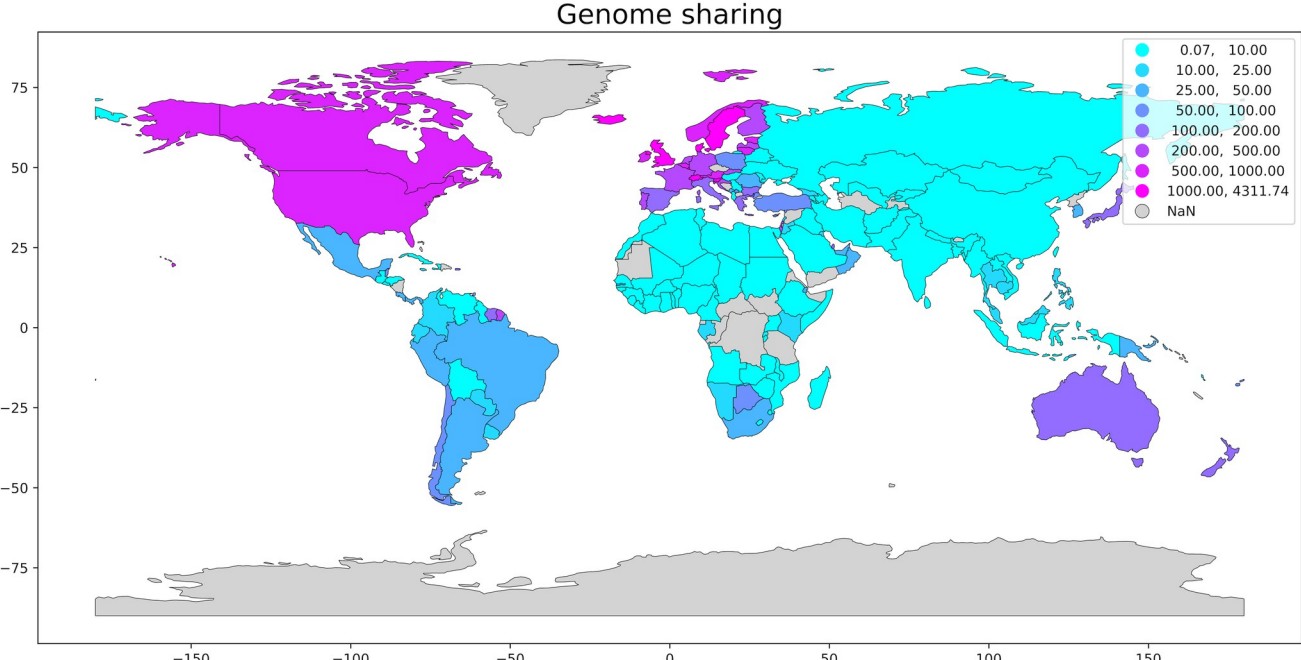

**Fig 3. Normalized distribution of the genome shared per $10^5$ inhabitants per country.** The areas in grey are without reported data. We have normalized the data to compare countries of different population sizes. The geospatial data for plotting the map was obtained in the open-source software library written for the Python programming language, Geopandas (source: GISAID platform).

**Table 1. Genome shared and lethality per country.**

| Country | Genome shared | Genome shared per $10^5$ inhabitants[§§] | Reported COVID-19 cases | Reported COVID-19 deaths | Lethality[§] |
|---|---|---|---|---|---|
| United States of America | 1,732,690 | 523 | 47,802,459 | 771,529 | 1.61% |
| United Kingdom | 1,282,315 | 1,889 | 10,021,501 | 144,433 | 1.44% |
| Germany | 260,519 | 311 | 5,650,170 | 100,476 | 1.78% |
| Denmark | 218,679 | 3,775 | 466,817 | 2.841 | 0.61% |
| Canada | 161,403 | 428 | 1,774,946 | 29,580 | 1,67% |
| France | 121,009 | 185 | 7,285,128 | 116,314 | 1.60% |
| India | 74,279 | 5 | 34,555,431 | 467,468 | 1.35% |
| Italy | 71,623 | 118 | 4,968,341 | 133,486 | 2.69% |
| Brazil | 75,292 | 35 | 22,043,112 | 613,339 | 2.78% |
| Mexico | 38,365 | 30 | 3,872,263 | 293,186 | 7.57% |
| South Africa | 23,634 | 40 | 2,952,500 | 89,771 | 3.04% |
| Russia | 9,982 | 7 | 9,502,879 | 270,292 | 2.84% |
| China | 1,203 | 0,1 | 127,631 | 5,697 | 4.46% |
| Iceland | 9,812 | 2,875 | 17,446 | 35 | 0.20% |

[§] [(Reported COVID-19 deaths)/(Reported COVID-19 cases)].

[§§]Genome sharing data were normalized per $10^5$ habitants to allow comparison between countries of rather different population sizes (source: GISAID platform). Data collected up to November 28th 2021.

aligned, and conserved regions were chosen using Clustal W program. An overlap of 100 nucleotides was programmed (Fig 4). Table 2 presents the sequence of primers used. Standard RT-PCR was performed using Superscript III one-step RT-PCR kit (Invitrogen, Carlsbad, CA, USA) according to the manufacturer's instructions, with 0.7 µM primers and temperature conditions according to Table 3.

The amplification of the fragments was visualized by 1,5% Agarose Gel Electrophoresis. The samples were quantified with the nanodrop one (Thermo Scientific™ NanoDrop™ One Microvolume UV-Vis Spectrophotometers) for sequencing.

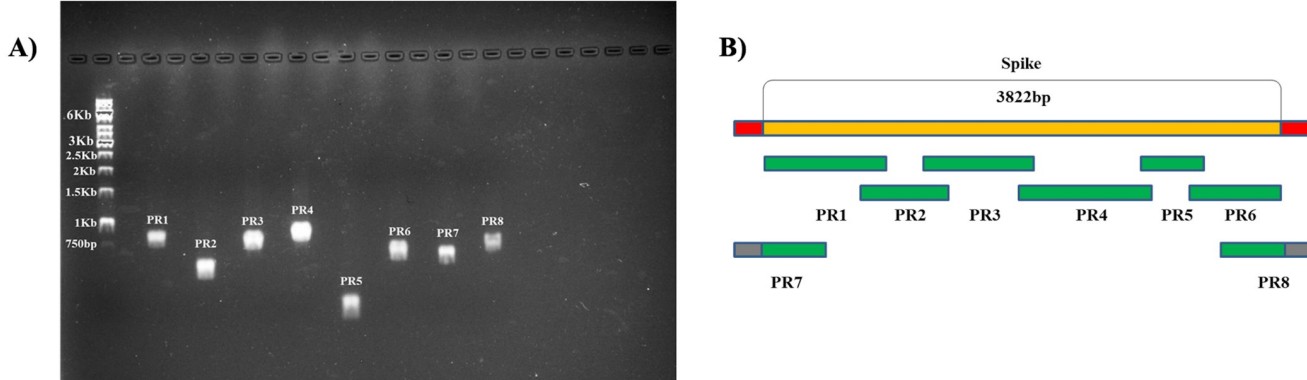

**Fig 4. Agarose Gel Electrophoresis and schematic representation of the targeted fragments of each set of primers.** Amplification of S gene is visualized in agarose Gel Electrophoresis (A). Schematic representation of the targeted fragments of each set of primers is shown in (B). PR1 represents primer set 1, PR2 primer set 2, PR3 primer set 3, PR4 primer set 4, PR5 primer set 5, PR6 primer set 6, PR7 primer set 7 and PR8 primer set 8.

**Table 2. List of primers used for the S gene amplification of SARS-CoV-2.**

| Primer S segment | Sequence (5'–3') | Coding Position | Product size (pb) |
|---|---|---|---|
| SP1 sense ASP1antisense | GTTTGTTTTTCTTGTTTTATT ACAGTGAAGGATTTCAACGTACAC | (21551–21574) (22450–22474) | 923pb |
| SP2 sense ASP2 antisense | CGTGATCTCCCTCAGGGTTTT TCAGCAATCTTTCCAGTTTGCC | (22190–22211) (22810–22832) | 620pb |
| SP3 sense ASP3 antisense | GTAATTAGAGGTGATGAAGTCAGA ACATAGTGTAGGCAATGATGGA | (22751–22775) (23621–23643) | 892pb |
| SP4 sense ASP4 antisense | CTTGGCGTGTTTATTCTACAG GCTTGTGCATTTTGGTTGACC | (23445–23466) (24403–24424) | 979pb |
| SP5 sense ASP5 antisense | AGACTCACTTTCTTCCACAGCA AGATGATAGCCCTTTCCACA | (24355–24377) (24699–24719) | 342pb |
| SP6 sense ASP6 antisense | TTCTGCTAATCTTGCTGCTACT GTTTATGTGTAATGTAATTTGACTCC | (24610–24632) (25348–25372) | 766pb |
| SP7 sense# ASP7 antisense | TAGAGAAAACAACAGAGTT TGAGGGAGATCACGCACTAA | (21492–21511) (22184–22204) | 712pb |
| SP8 sense# ASP8 antisense | TTCTGCTAATCTTGCTGCTACT CCTTGCTTCAAAGTTACAGTTCCA | (24610–24632) (25409–25433) | 825pb |

# Set primers 7 and 8 flanks the S protein-coding region.

Nucleotide positions are according to the SARS-CoV-2 Wuhan (Genbank accession no.NC_ 045512-Wuhan-HU-1).

## Nucleotide sequence determination and analysis

The nucleotide sequences were determined from 200 ng of the amplicon, using the Big Dye Terminator kit 3.1 (Applied Biosystems), following the manufacturer's procedure. Amplicons were sequenced in the ABI 3730 genetic analyzer (Applied Biosystems, USA) following the manufacturer's protocol. Raw sequence data were aligned, edited, assembled using the BioEdit Sequence Alignment Editor, Version 7.0.5.3.

The protocol described in this peer-reviewed article is published on protocols.io, https://dx.doi.org/10.17504/protocols.io.bx6kprcw and is included for printing as S1 File with this article.

## Results and discussion

This methodology covered 100% of the S gene sequenced (3,822 pb). The sequences obtained were deposited at GISAID numbers EPI_ISL_4496739, EPI_ISL_4497141, EPI_ISL_4497286.

All the eight primers set produced single amplicons for the three samples used to evaluate this protocol (Fig 4A); therefore, sequencing reaction could be performed without extracting

**Table 3. RT-PCR cycle conditions.**

| Temperature | Time | |
|---|---|---|
| 60˚C | 1 minute | Reverse transcription and Transcriptase inactivation |
| 50˚C | 45 minutes | |
| 94˚C | 2 minutes | |
| 95˚C | 15 seconds | 40 cycles of amplification |
| 53˚C | 30 seconds | |
| 68˚C | 1 minute | |
| 68˚C | 7 minutes | Final extension |

**Table 4. Protein S mutations of each VOC and studied sequences.**

| Sequences | AA identity | AA changes | Mutations | Accession number |
|---|---|---|---|---|
| Alpha (B.1.1.7) | 99.214% | 10 | H69del, V70del, Y144del, N501Y, A570D, D614G, P681H, T716I, S982A, D1118H | EPI_ISL_601443 |
| Beta (B.1.351) | 99.607% | 5 | D80A, E484K, N501Y, D614G, A701V | EPI_ISL_660613 |
| Gamma (P.1) | 99.057% | 12 | L18F, T20N, P26S, D138Y, R190S, K417T, E484K, N501Y, D614G, H655Y, T1027I, V1176F | \|EPI_ISL_906071 |
| Delta (B.1.617) | 99.214% | 10 | T19R, E156G, F157del, R158del, A222V, L452R, T478K, D614G, P681R, D950N | EPI_ISL_2047658 |
| Rio de Janeiro | 99.057% | 12 | L18F, T20N, P26S, D138Y, R190S, K417T, E484K, N501Y, D614G, H655Y, T1027I, V1176F | EPI_ISL_4496739 |
| Santo Antonio de Padua | 99.764% | 3 | E484K, D614G, V1176F | EPI_ISL_4497141 |
| Seropedica | 99.450% | 7 | E156D, E484K, D614G, D775V, T866P, M869K, V1176F | EPI_ISL_4497286 |

Nucleotide positions are according to the SARS-CoV-2 Wuhan (GenBank accession no. NC_ 045512-Wuhan-HU-1).

the bands from agarose gel. In addition, no mismatch in the primer regions that could lead to the escape of known VOCs was observed (S2 File).

The samples sequenced in this study originated from Rio de Janeiro City, Santo Antônio de Pádua and Seropédica, in Rio de Janeiro state. The obtained sequences were aligned with reference sequences of each VOC, in order to detect and compare mutations. Spike protein from Rio de Janeiro city sample displayed the same amino acid changes found in reference sequence of Gamma variant, suggesting that this sample is probably classified into P.1 lineage (Table 4). According to the literature, the P.1 lineage (gamma) emerged in Manaus, Amazonas, evolved from a B.1.1.28 clade in late November 2020 and replaced its parental lineage in less than two months[7]. We found a strain displaying similar spike protein with that of P.1 lineage circulating in Rio de Janeiro as early as February 2021.

The samples from Santo Antônio de Pádua and Seropédica didn't show similar mutation patterns with gamma VOC (Table 4), however, they presented some mutations of importance, like E484K and D614G (Fig 5). The change from glutamic acid to a lysine in the 484th amino acid position of the Spike protein (E484K) already occurred 228,871 times (4.27% of all samples with spike sequence) in 166 countries, according to GISAID Spike Glycoprotein Mutation

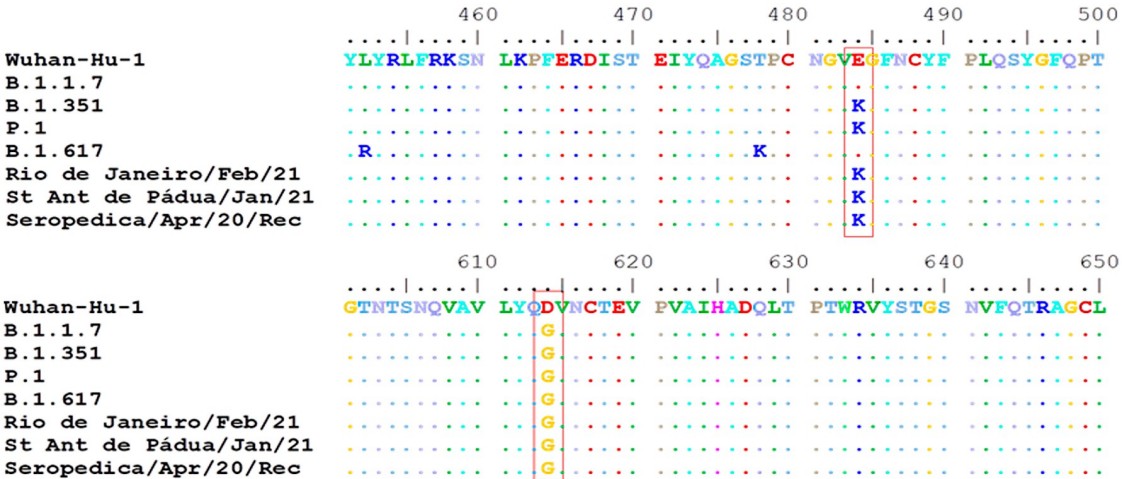

**Fig 5. Sequence alignment showing amino acid substitutions E484K and D614G.**

Surveillance. This mutation has been reported in the literature to be related to enhanced host receptor binding [8] and antigenic drift [9] either alone or in association with other mutations [10]. The mutation D614G is widely spread and has already occurred 5,285,437 times (98.51% of all samples with Spike sequence) in 204 countries. It was reported to be related to the increase in infectivity of SARS-CoV-2, higher viral loads, increased replication fitness, and virulence [11,12].

Apart from the mutations of high importance, the sequence from Seropédica also presented some rare mutations. The amino acid substitutions D775V, T866P and M869K are present in less than three sequences in GISAID database. The effects of these mutations are still unknown.

Due to its essential role in establishing infection, as well as inducing immune response, the genomic surveillance of the S protein of SARS-CoV-2 is of paramount importance. Monitoring the emergence of new variants, and the interactions between their mutations, allow the scientific community to develop better strategies to control the pandemic.

The count of genomic sequences obtained in each country reveals a vast disproportion that becomes evident in surveillance platforms like GISAID. One of the reasons for this disparity is the limited access to NGS methodologies by most groups. Therefore, this work describes a protocol for complete nucleotide sequencing of the S gene using the Sanger technique, which could be helpful to keep tracking SARS-CoV-2 protein S evolution.

## Supporting information

**S1 File. The PDF of the protocol described in this peer-reviewed article published on protocols.io dx.doi.org/10.17504/protocols.io.bx6kprcw.**
(PDF)

**S2 File. Map of primer pairs in VOCs sequences.**
(PDF)

**S3 File. Flowchart describing the sequential steps of the protocol.**
(PDF)

## Acknowledgments

We want to thank all health professionals, especially LACEN-RJ staff, for their collaboration during the implementation of this protocol and for all efforts in facing the COVID-19 pandemic.

## Author Contributions

**Conceptualization:** Andrea Cony Cavalcanti, Stefanella Boatto, Maria Angélica Arpon Marandino Guimarães, Renata Campos Azevedo.

**Formal analysis:** Tiago Souza Salles, Fábio Burack da Costa, Vanessa Zaquieu Dias, José Antônio Suzano da Silva.

**Funding acquisition:** Davis Fernandes Ferreira.

**Investigation:** Leandro Magalhães de Souza, Marcelo Damião Ferreira de Meneses, Cinthya Domingues Amaral, Jhonatan Ramos Felix, Duleide Alves Pereira.

**Project administration:** Renata Campos Azevedo.

**Supervision:** Stefanella Boatto, Maria Angélica Arpon Marandino Guimarães, Davis Fernandes Ferreira, Renata Campos Azevedo.

**Writing – original draft:** Tiago Souza Salles, Andrea Cony Cavalcanti, Fábio Burack da Costa, Vanessa Zaquieu Dias, José Antônio Suzano da Silva.

**Writing – review & editing:** Stefanella Boatto, Davis Fernandes Ferreira, Renata Campos Azevedo.

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
