## [Decision Letter · Decision Letter 0]

23 Nov 2021

PONE-D-21-34394Genomic surveillance of SARS-CoV-2 Spike protein by sanger sequencing.PLOS ONE

Dear Dr. Azevedo,

Thank you for submitting your manuscript to PLOS ONE. After careful consideration, we feel that it has merit but does not fully meet PLOS ONE’s publication criteria as it currently stands. Therefore, we invite you to submit a revised version of the manuscript that addresses the points raised during the review process.

The comments raised by the reviewers seem minor. Please provide your revised manuscript.

We look forward to receiving your revised manuscript.

Kind regards,

Etsuro Ito

Academic Editor

PLOS ONE

Journal Requirements:

2. Thank you for providing the following Protocols.io DOI in your submission form [Protocols.io DOI]. In keeping with our submission requirements, please add the Protocols.io DOI to the Methods section of your manuscript as well using this format: “The protocol described in this peer-reviewed article is published on protocols.io, https://dx.doi.org/10.17504/protocols.io[........] and is included for printing as supporting information file 1 with this article.” For more information, please see our submission guidelines: https://journals.plos.org/plosone/s/submission-guidelines#loc-guidelines-for-specific-study-types

3. Please ensure that you have specified (1) whether consent was informed, (2) what type you obtained (for instance, written or verbal, and if verbal, how it was documented and witnessed). If your study included minors, state whether you obtained consent from parents or guardians. If the need for consent was waived by the ethics committee and (3) If you are reporting a retrospective study of medical records or archived samples, please ensure that you have discussed whether all data were fully anonymized before you accessed them and/or whether the IRB or ethics committee waived the requirement for informed consent. If patients provided informed written consent to have data from their medical records used in research, please include this information.

 (The funders had and will not have a role in study design, data collection and analysis, decision to publish, or preparation of the manuscript.)

d) If you did not receive any funding for this study, please state: “The authors received no specific funding for this work.

(This work was supported by Fundação de Amparo à Pesquisa do Estado do Rio de Janeiro-FAPERJ [grant number E-26/201.840/2017] and Coordenação de Aperfeiçoamento de Pessoal de Nível Superior - Brasil (CAPES) – Public Notice Number 09/2020 - Prevention and Combat against Outbreaks, Endemics, Epidemics and Pandemics. Process number 223038.014313/2020-19, "Digital Technologies for Monitoring, Mapping, and Control of Outbreaks, Endemics, Epidemics and Pandemics", held at the Federal University of Rio de Janeiro (TSS and FBC) )

 (The funders had and will not have a role in study design, data collection and analysis, decision to publish, or preparation of the manuscript.)

Additionally, because some of your funding information pertains to [commercial funding//patents], we ask you to provide an updated Competing Interests statement, declaring all sources of commercial funding. 

In your Competing Interests statement, please confirm that your commercial funding does not alter your adherence to PLOS ONE Editorial policies and criteria by including the following statement: "This does not alter our adherence to PLOS ONE policies on sharing data and materials.” as detailed online in our guide for authors  http://journals.plos.org/plosone/s/competing-interests.  If this statement is not true and your adherence to PLOS policies on sharing data and materials is altered, please explain how. 

Please include the updated Competing Interests Statement and Funding Statement in your cover letter. We will change the online submission form on your behalf.

7. We note that you have included the phrase “data not shown” in your manuscript. Unfortunately, this does not meet our data sharing requirements. PLOS does not permit references to inaccessible data. We require that authors provide all relevant data within the paper, Supporting Information files, or in an acceptable, public repository. Please add a citation to support this phrase or upload the data that corresponds with these findings to a stable repository (such as Figshare or Dryad) and provide and URLs, DOIs, or accession numbers that may be used to access these data. Or, if the data are not a core part of the research being presented in your study, we ask that you remove the phrase that refers to these data.

8. We note that Figures 2 and 3 in your submission contain [map/satellite] images which may be copyrighted. All PLOS content is published under the Creative Commons Attribution License (CC BY 4.0), which means that the manuscript, images, and Supporting Information files will be freely available online, and any third party is permitted to access, download, copy, distribute, and use these materials in any way, even commercially, with proper attribution. For these reasons, we cannot publish previously copyrighted maps or satellite images created using proprietary data, such as Google software (Google Maps, Street View, and Earth). For more information, see our copyright guidelines: http://journals.plos.org/plosone/s/licenses-and-copyright.

A. You may seek permission from the original copyright holder of Figures 2 and 3 to publish the content specifically under the CC BY 4.0 license.  

B. If you are unable to obtain permission from the original copyright holder to publish these figures under the CC BY 4.0 license or if the copyright holder’s requirements are incompatible with the CC BY 4.0 license, please either i) remove the figure or ii) supply a replacement figure that complies with the CC BY 4.0 license. Please check copyright information on all replacement figures and update the figure caption with source information. If applicable, please specify in the figure caption text when a figure is similar but not identical to the original image and is therefore for illustrative purposes only.

Reviewers' comments:

Reviewer's Responses to Questions

**Comments to the Author**

1. Does the manuscript report a protocol which is of utility to the research community and adds value to the published literature?

Reviewer #1: Yes

Reviewer #2: Yes

2. Has the protocol been described in sufficient detail?

Descriptions of methods and reagents contained in the step-by-step protocol should be reported in sufficient detail for another researcher to reproduce all experiments and analyses. The protocol should describe the appropriate controls, sample sizes and replication needed to ensure that the data are robust and reproducible.

Reviewer #1: Yes

Reviewer #2: Yes

3. Does the protocol describe a validated method?

Reviewer #1: Yes

Reviewer #2: No

4. If the manuscript contains new data, have the authors made this data fully available?

Reviewer #1: Yes

Reviewer #2: Yes

**5. Is the article presented in an intelligible fashion and written in standard English?**

Reviewer #1: Yes

Reviewer #2: Yes

6. Review Comments to the Author

Reviewer #1: In the study of Salles and Cavalcanti et al., entitled “Genomic surveillance of SARS-CoV-2 Spike gene by sanger sequencing” the authors describe a new feasible protocol for complete nucleotide sequencing of the Spike (S) gene using the Sanger technique. Thus, the authors conclude that such a methodology could be easily adopted by any laboratory with experience in sequencing, adding to effective surveillance of SARS-CoV-2 spreading and evolution. The study is important and should be accepted after minor revisions.

Minor revisions:

1) Change title to “…SARS-CoV-2 Spike gene…” (and not protein);

2) Page 7, line 25: change to “…mainly in Spike (S) gene. Monitoring the S gene” (and not protein);

3) Page 7, line 29: change to “…SARS-CoV-2 S gene” (and not protein);

4) Page 7, line 30: change to “…S gene” (and not protein);

5) Page 7, line 33: change to “…Spike gene” (and not protein);

6) Page 7, line 34: change to “…SARS-CoV-2 Spike gene” (and not protein);

7) Page 8, line 41: Describe the acronym: “… been denominated severe acute respiratory syndrome coronavirus 2 (SARS‐CoV‐2).”;

8) Page 9, lines 86 and 87: change to: “Unlike NGS methodologies, nucleotide sequencing based on the Sanger technique is widespread worldwide.”;

9) Page 9, lines 88 and 89: change to “…SARS-CoV-2 S gene” and “S gene” (and not protein);

10) Page 10, line 133: change to “S gene” (and not protein);

11) Page 10, line 139: remove (data not shown);

12) Page 11, lines 152 and 156: update the number of times these mutations appear in GISAID;

13) Page 11, line 171: change to “S gene” (and not protein);

14) Page 14, line 250: update collected data;

15) Page 14, line 251 (Table 2): I suggest changing the primers’ names so they are not confused with the P1 (Gamma) variant and its sublineages;

16) Page 16, line 275: change to “…S gene” (and not protein);

17) Page 16, line 277: change to “…S gene” (and not protein).

Reviewer #2: The manuscript entitled "Genomic surveillance of SARS-CoV-2 Spike protein by Sanger sequencing" presents a protocol for complete sequencing of the S protein of SARS-CoV-2 using the Sanger technique, as an alternative to next-generation sequencing (NGS), in the investigation of mutations in the S protein of SARS-CoV-2 at lower cost.

The manuscript presents the steps of the protocol in a complete and detailed manner.

I recommend that a flowchart with the sequential steps of the proposed protocol be inserted.

lines 117 - 119 - I recommend that the data referring to cycling be presented in the form of a table, in order to facilitate understanding.

line 136 - I recommend replacing the term validate by evaluate

7. PLOS authors have the option to publish the peer review history of their article (what does this mean?). If published, this will include your full peer review and any attached files.

Reviewer #1: **Yes: **Fabrício Souza Campos

Reviewer #2: No

---

## [Author Response · Author response to Decision Letter 0]

10 Dec 2021

Response to Reviewers

Dear Etsuro Ito, 

We want to thank you for the comments. The manuscript was revised carefully, attending to the points raised during the review process. We hope that this version fully meets PLOS ONE’s publication criteria. 

Kind regards, 

Renata Campos Azevedo

Journal Requirements:

R. The manuscript was revised according to style requirements.

2. Thank you for providing the following Protocols.io DOI in your submission form [Protocols.io DOI]. In keeping with our submission requirements, please add the Protocols.io DOI to the Methods section of your manuscript as well using this format: “The protocol described in this peer-reviewed article is published on protocols.io, https://dx.doi.org/10.17504/protocols.io[........] and is included for printing as supporting information file 1 with this article.” For more information, please see our submission guidelines: https://journals.plos.org/plosone/s/submission-guidelines#loc-guidelines-for-specific-study-types

R. The protocols.io DOI was added to the methods section using the format described (lines 227 up to 229 from the revised version). “The protocol described in this peer-reviewed article is published on protocols.io, https://dx.doi.org/10.17504/protocols.io.bx6kprcw and is included for printing as supporting information file 1 with this article.”

3. Please ensure that you have specified (1) whether consent was informed, (2) what type you obtained (for instance, written or verbal, and if verbal, how it was documented and witnessed). If the need for consent was waived by the ethics committee and (3) If you are reporting a retrospective study of medical records or archived samples, please ensure that you have discussed whether all data were fully anonymized before you accessed them and/or whether the IRB or ethics committee waived the requirement for informed consent.

R. The information regarding the waive of consent by the ethics committee was included (lines 134 up to 137). “Human samples were used after the conclusion of the diagnostic investigation. All patients' personal information was anonymized, only the municipalities of residence were disclosed. Therefore, the ethics committee waived the requirement for informed consent from patients.”

Financial disclosure:

R. The financial disclosure was corrected.

This work was material supported by Fundação Carlos Chagas Filho de Amparo à Pesquisa do Estado do Rio de Janeiro-FAPERJ [grant number E-26/201.840/2017] (RCA) and Coordenação de Aperfeiçoamento de Pessoal de Nível Superior - Brasil (CAPES) – Public Notice Number 09/2020 - Prevention and Combat against Outbreaks, Endemics, Epidemics and Pandemics. Process number 223038.014313/2020-19 (TSS and FBC). The funders had no role in study design, data collection and analysis, decision to publish, or preparation of the manuscript. 

Additionally, because some of your funding information pertains to [commercial funding//patents], we ask you to provide an updated Competing Interests statement, declaring all sources of commercial funding. In your Competing Interests statement, please confirm that your commercial funding does not alter your adherence to PLOS ONE Editorial policies and criteria by including the following statement: "This does not alter our adherence to PLOS ONE policies on sharing data and materials.” as detailed online in our guide for authors http://journals.plos.org/plosone/s/competing-interests. If this statement is not true and your adherence to PLOS policies on sharing data and materials is altered, please explain how.

R. Our research grant was from public organizations and not from commercial funding//patents. Therefore, the authors reaffirm that no competing interests exist.

7. We note that you have included the phrase “data not shown” in your manuscript. Unfortunately, this does not meet our data sharing requirements. PLOS does not permit references to inaccessible data. We require that authors provide all relevant data within the paper, Supporting Information files, or in an acceptable, public repository. Please add a citation to support this phrase or upload the data that corresponds with these findings to a stable repository (such as Figshare or Dryad) and provide and URLs, DOIs, or accession numbers that may be used to access these data. Or, if the data are not a core part of the research being presented in your study, we ask that you remove the phrase that refers to these data.

R. The data was included as supporting Information file 2. 

8. We note that Figures 2 and 3 in your submission contain [map/satellite] images which may be copyrighted. All PLOS content is published under the Creative Commons Attribution License (CC BY 4.0), which means that the manuscript, images, and Supporting Information files will be freely available online, and any third party is permitted to access, download, copy, distribute, and use these materials in any way, even commercially, with proper attribution. For these reasons, we cannot publish previously copyrighted maps or satellite images created using proprietary data, such as Google software (Google Maps, Street View, and Earth). For more information, see our copyright guidelines: http://journals.plos.org/plosone/s/licenses-and-copyright.

R. We want to clarify that both Figures 2 and 3 were made originally for this article using the GISAID platform's data to estimate COVID Lethality and genome sharing per 105 inhabitants for various countries around the world, as cited in the figure caption. We obtained the geospatial data for plotting the map in the open-source software library written for the Python programming language, Geopandas. Like any open source project, Geopanda and its internal datasets are free for all to use and released under the liberal terms of the BSD-3-Clause license. We added Geopandas' reference to the bibliography to give its proper credit as a fundamental tool for plotting these maps.

R. All references were checked, and no retracted article was cited.

Please find the answers to reviewers’ specific comments

Reviewer #1: In the study of Salles and Cavalcanti et al., entitled “Genomic surveillance of SARS-CoV-2 Spike gene by sanger sequencing” the authors describe a new feasible protocol for complete nucleotide sequencing of the Spike (S) gene using the Sanger technique. Thus, the authors conclude that such a methodology could be easily adopted by any laboratory with experience in sequencing, adding to effective surveillance of SARS-CoV-2 spreading and evolution. The study is important and should be accepted after minor revisions.

Minor revisions:

1) Change title to “…SARS-CoV-2 Spike gene…” (and not protein); 

2) Page 7, line 25: change to “…mainly in Spike (S) gene. Monitoring the S gene” (and not protein);

3) Page 7, line 29: change to “…SARS-CoV-2 S gene” (and not protein);

4) Page 7, line 30: change to “…S gene” (and not protein);

5) Page 7, line 33: change to “…Spike gene” (and not protein);

6) Page 7, line 34: change to “…SARS-CoV-2 Spike gene” (and not protein);

9) Page 9, lines 88 and 89: change to “…SARS-CoV-2 S gene” and “S gene” (and not protein);

10) Page 10, line 133: change to “S gene” (and not protein);

13) Page 11, line 171: change to “S gene” (and not protein);

16) Page 16, line 275: change to “…S gene” (and not protein);

17) Page 16, line 277: change to “…S gene” (and not protein).

R: The manuscript was revised, and all phrases containing protein S written referring to gene S were correct. 

7) Page 8, line 41: Describe the acronym: “… been denominated severe acute respiratory syndrome coronavirus 2 (SARS‐CoV‐2).”;

R: The acronym was described.

8) Page 9, lines 86 and 87: change to: “Unlike NGS methodologies, nucleotide sequencing based on the Sanger technique is widespread worldwide.”;

R: The term worldwide was used. 

11) Page 10, line 139: remove (data not shown);

R: The statement was removed, and the data was included as supporting Information file 2.

12) Page 11, lines 152 and 156: update the number of times these mutations appear in GISAID;

R: The number of times the mutations appear was updated (lines 267 up to 277, revised version)

14) Page 14, line 250: update collected data;

R:The data was updated up to November 28th, 2021, from the table and maps.

15) Page 14, line 251 (Table 2): I suggest changing the primers’ names so they are not confused with the P1 (Gamma) variant and its sublineages;

R: Thank you for your suggestion. Sense primers are now named SP 1 up to 8 and antisense ASP 1 up to 8. Primers pairs was named as PR 1 up to 8

Reviewer #2: The manuscript entitled "Genomic surveillance of SARS-CoV-2 Spike protein by Sanger sequencing" presents a protocol for complete sequencing of the S protein of SARS-CoV-2 using the Sanger technique, as an alternative to next-generation sequencing (NGS), in the investigation of mutations in the S protein of SARS-CoV-2 at lower cost. The manuscript presents the steps of the protocol in a complete and detailed manner.

I recommend that a flowchart with the sequential steps of the proposed protocol be inserted.

R: The flowchart describing the sequential steps was included as supporting information file 3

lines 117 - 119 - I recommend that the data referring to cycling be presented in the form of a table, in order to facilitate understanding.

R: A table including the RT-PCR cycling was included in the Materials & Methods section.

line 136 - I recommend replacing the term validate by evaluate

R: The term validate was replaced as the suggestion (lines 130 and 237, revised version).

---

## [Editor Report · Decision Letter 1]

17 Dec 2021

Genomic surveillance of SARS-CoV-2 Spike gene by sanger sequencing.

PONE-D-21-34394R1

Dear Dr. Azevedo,

We’re pleased to inform you that your manuscript has been judged scientifically suitable for publication and will be formally accepted for publication once it meets all outstanding technical requirements.

Kind regards,

Etsuro Ito

Academic Editor

PLOS ONE

---

## [Editor Report · Acceptance letter]

11 Jan 2022

PONE-D-21-34394R1 

Genomic surveillance of SARS-CoV-2 Spike gene by sanger sequencing.

Dear Dr. Azevedo:

I'm pleased to inform you that your manuscript has been deemed suitable for publication in PLOS ONE. Congratulations! Your manuscript is now with our production department. 

Kind regards, 

on behalf of

Prof. Etsuro Ito 

Academic Editor

PLOS ONE